# Optical, Structural, and Dielectric Properties of Composites Based on Thermoplastic Polymers of the Polyolefin and Polyurethane Type and BaTiO_3_ Nanoparticles

**DOI:** 10.3390/ma14040753

**Published:** 2021-02-05

**Authors:** M. Baibarac, A. Nila, I. Smaranda, M. Stroe, L. Stingescu, M. Cristea, R. C. Cercel, A. Lorinczi, P. Ganea, I. Mercioniu, R. Ciobanu, C. Schreiner, R. G. Garcia, C. Bartha

**Affiliations:** 1National Institute of Materials Physics, Laboratory of Optical Processes in Nanostructured Materials, Atomistilor Street 405A, P.O. Box MG-7, R077125 Bucharest, Romania; andreea.nila@infim.ro (A.N.); ion.smaranda@infim.ro (I.S.); malvina@infim.ro (M.S.); luiza.stingescu@infim.ro (L.S.); mirela.cristea@infim.ro (M.C.); radu.cercel@infim.ro (R.C.C.); lorinczi@infim.ro (A.L.); paul.ganea@infim.ro (P.G.); 2National Institute of Materials Physics, Atomic Structures and Defects in Advanced Materials Laboratory, Atomistilor Street 405A, P.O. Box MG-7, R077125 Bucharest, Romania; imercioniu@infim.ro; 3SC All Green SRL, 8 George Cosbuc, 700470 Iasi, Romania; rciobanu@yahoo.com (R.C.); cschrein@ee.tuiasi.ro (C.S.); 4Faculty of Electrical Engineering, Department of Electrical Measurements and Materials, Technical University Gh. Asachi Iasi, Bd. Professor Dimitrie Mangeron 67, 70050 Iasi, Romania; 5Izertis, Parque Cientifico Tecnologico, Avda. Del Jardin Botanico, 1345 Edificio Intra, 33203 Gijon, Spain; raquel.garcia@izertis.com; 6National Institute of Materials Physics, Magnetism and Superconductivity Laboratory, Atomistilor Street 405A, P.O. Box MG-7, R077125 Bucharest, Romania; cristina.bartha@infim.ro

**Keywords:** thermoplastic polymers, BaTiO_3_, crystallization processes

## Abstract

In this work, new films containing composite materials based on blends of thermoplastic polymers of the polyurethane (TPU) and polyolefin (TPO) type, in the absence and presence of BaTiO_3_ nanoparticles (NPs) with the size smaller 100 nm, were prepared. The vibrational properties of the free films depending on the weight ratio of the two thermoplastic polymers were studied. Our results demonstrate that these films are optically active, with strong, broad, and adjustable photoluminescence by varying the amount of TPU. The crystalline structure of BaTiO_3_ and the influence of thermoplastic polymers on the crystallization process of these inorganic NPs were determined by X-ray diffraction (XRD) studies. The vibrational changes induced in the thermoplastic polymer’s matrix of the BaTiO_3_ NPs were showcased by Raman scattering and FTIR spectroscopy. The incorporation of BaTiO_3_ NPs in the matrix of thermoplastic elastomers revealed the shift dependence of the photoluminescence (PL) band depending on the BaTiO_3_ NP concentration, which was capable of covering a wide visible spectral range. The dependencies of the dielectric relaxation phenomena with the weight of BaTiO_3_ NPs in thermoplastic polymers blends were also demonstrated.

## 1. Introduction

Thermoplastic polyurethane (TPU) is a linear elastomer with both non-polar soft and polar hard segments, resulting from the polymerization reaction of diisocyanates having diols with short- and long-chains, respectively [1,2]. The main interest in TPU and its nanocomposites lies in the fabrication of fused filaments, electronic devices, automotive panels, sporting goods, belts, profile, tubes, hoses, and so on [2,3,4]. Depending on the soft segments, polyester- or polyether-based TPUs are differentiated by abrasion resistance, mechanical properties, hydrolysis resistance, heat aging, injectability, and so on [5,6,7]. The heat resistance as well as the stiffness of TPU are relatively low and therefore blending them with other polymer matrices is seen as an attractive way to improve these properties [8,9]. In this context, blends of TPU with insulating macromolecular compounds such as polyamide [10], polylactic acid [11], polycaprolactone [12], and polyolefins [13,14] have been intensively studied. Thermoplastic polyolefin (TPO), known as polyethylene-polyoctene rubbers, is one of the most common and commercial thermoplastic polymers used due to its high rigidity and good chemical resistance [15,16]. In TPU:TPO mixtures, the polarity and fragility of the TPO are improved, while the low thermal stability, processability, and some mechanical properties of TPU are considerably enhanced [17,18,19].

Since 2013, a continuous effort has been devoted in the development of new composites based on BaTiO_3_ NPs, thermoplastic polymers such as polyvinylidene fluoride (PVDF), and various carbon NPs such as graphene [20] or carbon nanotubes [21] for applications in the field of 3D printing devices [3,22], medical devices [21], and textile fibers [21].

As is well known, BaTiO_3_ is characterized by a high permittivity of about 10^5^ and a low dielectric loss of ~0.05 [23]. Due to the low polarization ability and high dielectric loss of thermoplastic elastomers, the incorporation of a highly polarizable filler like BaTiO_3_ NPs in the thermoplastic polymer matrix promotes interfacial (exchange coupling) effects that could significantly enhance the final dielectric performance of the resulting blends for specific electro-active applications [20]. 

The most widely used solvent mixing techniques for BaTiO_3_ nanocomposites are based on the solution casting method or in situ polymerization [20]. At the industrial level, the improvement of the manufacturing performance of polymer blends has led to a melting mixing technique using a co-rotating twin screw extruder for the fabrication of BaTiO_3_—elastomer nanocomposites [24]. In this context, we note that it has already found a high real dielectric permittivity (ε_r_ = 5.06) and a low dielectric loss (less than 0.05) in the case of BaTiO_3_ incorporated poly(styrene-ethylene/butylene-styrene)-grafted-maleic anhydride elastomers [24] and a significant dielectric permittivity at low frequency (values exceeding 10^4^) in the PVDF/BaTiO_3_ composites [20].

The realization of applications presupposes a good knowledge of the crystallization process of BaTiO_3_ in the PVDF matrix. [25] In the case of PVDF/BaTiO_3_ composites, the addition of inorganic particles to the matrix of the macromolecular compound has been demonstrated to induce a change in the interfacial polarization process and in the crystallization kinetics of heterogenous nucleation when a wider range of polymer melting temperatures in the presence of BaTiO_3_ NPs has been reported [20,25]. Knowledge of the crystallization process of BaTiO_3_ NPs in the matrix of TPU and TPO thermoplastic polymers can open perspectives for new applications such as for 3D and 4D printed devices [4,26]. Therefore, preliminary studies focused on the optical, structural, and dielectric properties of composites based on TPU:TPO blends and BaTiO_3_ NPs will be shown in this work.

## 2. Materials and Methods

### 2.1. Materials

Thermoplastic elastomers (i.e., TPU and TPO including a hardener, which corresponds to polydimethylsiloxane, marketed under the name of Sylgard TM 186 Silicone elastomer curing agent) were purchased from the Elastollan-BASF Chemical Company (Cleveland, OH, USA). BaTiO_3_ nanopowder, N,N′-dimethyl formamide (DMF), and C_2_H_5_OH were purchased from Sigma Aldrich (St. Louis, MO, USA).

### 2.2. The Preparation of TPU:TPO Films in the Absence and Presence of BaTiO_3_ NPs

The TPU:TPO blends were prepared as free films as follows. In the first stage, a solution of TPU in dimethylformamide (DMF) (0.5 g/8 mL) and another of TPO in C_2_H_5_OH (0.5 g/10 mL), to which was added 0.1 g of the TPO hardener in 2 mL DMF, were prepared under ultrasonication. After ultrasonically mixing them for 5 min, the resulting solution was placed in a Petri dish and dried for 2 h at 100 °C. The resulting film was peeled off the Petri glass and labeled as a TPU:TPO 1:1 blend, this being dried under vacuum to constant weight. Using this protocol and changing the mass ratio of the two thermoplastic polymers, five other free films labeled as blends of TPU:TPO 1:3, TPU:TPO 1:2, TPU:TO 2:1, TPU:TPO 4:1, and TPU:TPO 6:1 were prepared. The thickness of all blends was ca. 43 µm. 

The composites based on the TPU:TPO 1:1 blends and BaTiO_3_ NPs were prepared as described above, with the only difference being that after mixing the TPU and TPO solutions, the various weights of the inorganic particles (i.e., 0.05, 0.1, 0.23, and 0.3 g) were added under ultrasonication for 10 min. After the thermal treatment at 100 °C and drying under vacuum to constant weight, films labeled as composites based on TPU:TPO 1:1 and BaTiO_3_, having the concentration of inorganic NPs equal to 6.25, 12, 25, and 30 wt.%, respectively, were obtained. As the BaTiO_3_ NP concentration in the polymer weight increased from 6.25, 12, 25, and 30 wt.%, the thicknesses of the films changed to ~43.4, 44.0, 44.6m and 45 µm, respectively.

### 2.3. Microscopy Analysis 

Scanning electron microscopy (SEM) images of BaTiO_3_ NPs and the composites with the PTU:TPO blends were recorded with a scanning electron microscope, model Tescan Lyra III XMU (Libušina tř. 21 623 00, Brno—Kohoutovice, Czech Republic).

BaTiO_3_ NPs were studied by high-resolution transmission electron microscopy (HRTEM) (JEOL Ltd., Tokyo, Japan). In this order, a suspension of BaTiO_3_ NPs in C_2_H_5_OH was prepared and successively transferred onto a copper grid coated with an amorphous carbon support. HRTEM images of the BaTiO_3_ NPs were recorded with a JEOL JEM ARM 200 F electron microscope working at 200 keV.

### 2.4. X-ray Diffraction Analysis 

X-ray diffraction (XRD) patterns of TPU, TPO, BaTiO_3_, and their composites were performed in a theta–theta configuration in the angular range of 2θ = 5°–65° with a Bruker D8 Advance diffractometer (Bruker, Hamburg, Germany).

### 2.5. Fourier Transform Infrared (FTIR) Spectroscopic Analysis 

The IR spectra of TPU, TPO, BaTiO_3_, and their composites were recorded using a FTIR spectrophotometer, Vertex 80 model from Bruker (Billerica, MA, USA) with a resolution of 2 cm^−1^.

### 2.6. FT-Raman Spectroscopic Analysis 

The Raman spectra of TPU, TPO, BaTiO_3_, and their composites were recorded with a FT-Raman spectrophotometer, RFS 100S model, from Bruker (Ettlingen, Germany) with a resolution of 1 cm^−1^.

### 2.7. Photoluminescence Analysis 

The photoluminescence (PL) spectra of TPU, TPO, BaTiO_3_, and their composites were recorded with a Fluorolog-3 spectrophotometer, FL3-2.2.1 model, from Horiba Jobin Yvon (Palaiseau, France).

### 2.8. Dielectric Properties

The dielectric properties of composites based on the TPU:TPO blends and BaTiO_3_ NPs were examined using dielectric spectroscopy (DS) with the equipment having a high resolution Alpha-A Analyzer from NOVOCONTROL GmBH. The electrical properties were determined at room temperature with the frequency being between 0.01 Hz and 10 MHz and the alternating voltage having the value of 0.3 V. The samples were in the form of platelets with a diameter of approximately 13 mm and their thicknesses varied between 0.095 mm and 0.160 mm. The samples were placed between two circular metal electrodes in a capacitor configuration with plane-parallel armatures.

### 2.9. Differential Scanning Calorimetry (DSC) Analysis 

Differential scanning calorimetry (DSC) measurements were performed with a DSC 204 F1 type from Netzsch (Selb, Germany). The experiments were conducted at room temperature to 500 °C in an inert atmosphere of helium (40 mL/min flow gaze rate) and a heating rate of 5 °C/min. The samples were sealed by pressing and an empty Al crucible= was used as the reference. The accuracy of the heat flow measurements was ±0.001 mW.

## 3. Results and Discussions

### 3.1. Morphological and Structural Properties of the Composites Based on the TPU:TPO Blends and BaTiO_3_ NPs

The SEM micrograph of the films based on the TPU:TPO 1:1 blends and the BaTiO_3_ NPs are shown in Figure 1. At present, it is necessary to note that TPO is a material that combines crystalline or semi-crystalline thermoplastic components made of polyolefins (hard domain) with amorphous elastomeric components of polyethylene-polyoctene rubbers (soft structure) [27]. Figure 1a reveals a complex morphology consisting of continuous phases of TPO and TPU films, accompanied by a discrete phase that is randomly distributed. Taking into account the TPU composition, it is composed of hard polar segments (urethane component) and soft less-polar segments (long-chain diol) with glassy and rubbery characteristics, respectively. Such a structure is described by a high and low glass transition temperature that induces the thermodynamic incompatibility between hard and soft segments. In this context, the self-assembly of hard segments through physical bonds causes the Gibbs free energy of the film to become positive, inducing a two-phase separation with the formation of a hard-micro-domain aggregation of the urethane segments [28]. Simultaneously, a discrete phase is expected to appear in the TPO film due to a crystalline form of isotactic polypropylene, most often used as the polyolefin thermoplastic phase in the TPO composition [27]. Consequently, the discrete phases of approximately spherical shape belong to both TPO and TPU and they are randomly distributed to form the cross-physical contacts between the amorphous segments of TPU and TPO. A uniform dispersion of BaTiO_3_ NPs in the structure of thermoplastic polymers is observed as the amount of BaTiO_3_ increases. Most likely, the urethane polar group of TPU ensures good interfacial adhesion and a better dispersion of the polarizable microparticles of BaTiO_3_ in the TPU:TPO blends with the increase in BaTiO_3_ concentration.

In order to show additional information concerning the size of the inorganic NPs, Figure 2 shows the HRTEM image of the BaTiO_3_ NPs. The analysis of Figure 2 indicates that in the case of the particle count equal to 100, values of the minimum, maximum, and mean size of the BaTiO_3_ NPs equaled 30.04 ± 9.89 nm, 79.93 ± 9.89 nm, and 53.35 ± 9.89 nm, respectively.

The XRD spectra of BaTiO_3_, TPU, TPO, and their composites are shown in Figure 3. The XRD pattern of BaTiO_3_ is shown in Figure 3a, and revealed the crystalline tetragonal phase of BaTiO_3_ according to the standard International Centre for Diffraction Data (ICCD) database (PDF 04-016-2042) and the main diffraction peaks appeared at 2θ equal to 22.2°, 31.5°, 38.9°, 45.4°, 51°, and 56.3°, these being assigned to the crystalline planes (001), (110), (111), (002), (210), and (211), respectively. The XRD patterns of the two thermoplastic polymers (i.e., TPO and TPU) presented an amorphous structure highlighted by a non-crystalline board peak having the maximum 2θ equal to 19.9° (Figure 3b) and 19.6° (Figure 3c), respectively. In addition, Figure 3b shows a series of small maxima, most probably corresponding to a minor ordered phase formed inside the TPU polymer. The most prominent maximum from this series was the first one, at around 29.3°, followed by smaller ones at 35.9°, 39.3°, 43.1°, and 47.3°. These data correlated rather well with a TPU diffraction pattern reported in [29]. In contrast with TPU, significant differences were observed for the XRD pattern of TPO in the 2θ angular domain between 40° and 65°. The presence of peaks from 2θ equal to 44.3° and 64.5° in the XRD pattern of TPO have also been reported by Z.J. Zhang et al. [30]. These can be correlated with both propylene chains of TPO (hard segments) and carbon black, which is incorporated in the TPO fabrication as an additive in order to stabilize the membrane against UV irradiation or as a filler to increase the hardness of the membrane [31].

As expected, the XRD pattern of the TPU:TPO 2:1 blend (Figure 3d) contained all the peaks of the two thermoplastic polymers. Figure 3e shows the XRD pattern of the composite based on the TPU:TPO 2:1 blend and BaTiO_3_ NPs when the inorganic compound concentration was equal to 30 wt.%. In this last case, a sum effect of the three constituents, without disturbing the tetragonal structure of the BaTiO_3_ NPs, can be remarked.

### 3.2. Vibrational Properties of the Composites Based on the TPU:TPO Blends and BaTiO_3_ NPs

Figure 4 shows the Raman spectra of films based on the TPU and TPO thermoplastic polymers as well as their blends. According to Figure 4, the Raman spectrum of TPO can be characterized by lines located at 2908–2968 cm^−1^, assigned to the C–H stretching vibrational mode [32]. In the case of the TPU:TPO blend, Figure 4 highlights the following Raman lines located at 1300, 1440, 1618, and 2864 cm^−1^, which are attributed to the vibrational modes of deformation of CH–urethane amide, symmetrical stretch of N=C=O + CH_2_ deformation, aromatic stretching structure, and CH stretching in the aromatic structure, respectively [33,34].

Increasing the TPU weight in the films based on the TPU:TPO blends led to: (i) a gradual shift of the Raman line belonging to TPO from 2908 cm^−1^ (Figure 4a) to 2912 cm^−1^ (Figure 4e), 2914 cm^−1^ (Figure 4f), 2916 cm^−1^ (Figure 4g), and 2918 cm^−1^ (Figure 4b); and (ii) a change in the ratio between the relative intensities of the Raman lines located at 1616 cm^−1^ and 2912–2916 cm^−1^ in favor of the first line. Figure 5e highlights that the main Raman lines of the BaTiO_3_ peaked at 268, 308, 520, and 715 cm^−1^, these being assigned to the vibrational modes A_1_ (TO), E (TO + LO), A_1_ (TO), and E (LO), respectively [35].

According to Figure 5, depending on the BaTiO_3_ concentration in the film containing the TPU:TPO 2:1 blends, we observed: (i) a gradual increase in the intensity of the Raman lines related to the guest inorganic particle as their concentration increased; (ii) a change in the position of the Raman line from 520 cm^−1^ (Figure 5e) to 513 cm^−1^ (Figure 5a) at a lower concentration of BaTiO_3_ in the TPU:TPO 2:1 blend (6 wt.%); (iii) an up-shift of the Raman line from 1300 cm^−1^ (Figure 4b) to 1307 cm^−1^ (Figure 5d) as the BaTiO_3_ concentration increased in the films’ structure; and (iv) a down-shift of the Raman line at 2918 cm^−1^ (Figure 4b) to 2910 cm^−1^ (Figure 5d). These changes suggest an adsorption of the thermoplastic polymers on the surface of BaTiO_3_ NPs via C–H bonds of the TPU amide groups. Complementary information on the vibrational properties of BaTiO_3_, TPU, TPO, and their composites are presented by IR spectroscopy. According to Figure 6, the main IR absorption bands of TPO are located at 787, 1009, 1258, and 2963 cm^−1^ (Figure 6a); the absorption band of 787 cm^−1^ can be attributed to the complex vibration mode of CH_2_ rock + C-C chain stretching + CH bending [36], and the bands of 1009 and 1258 cm^−1^ correspond to the –CH_3_ rocking vibrational modes and CH bending vibrations, respectively [37].

The last absorption band at 1258 cm^−1^ can be assigned to the asymmetric stretching vibration modes of the CH_3_- groups [37] and the stretching vibrations of the C–H bonds [32]. The TPUs are located at 816, 1076–1103, 1221, 1310, 1414, 1530, 1597, 1701–1730, 2853–2938, and 3300–3728 cm^−1^ (Figure 6b). According to the studies reported thus far, the IR band of bending vibrational modes of N–H can be observed at 816 cm^−1^, while the absorption bands from 1000–1150, 1202, 1597, and 3100–3600 cm^−1^ are attributed to the vibrational modes of stretching C(O)–OC, CO stretching in the ether group, C-C + C=C in the benzene ring, and the stretching vibrational modes of the NH group, respectively [38,39,40]. The IR bands of 2853 and 2938 cm^−1^ are attributed to the anti-symmetrical and symmetrical vibrational modes of the CH bonds, respectively [41,42]. The absorption band located at 1701 cm^−1^ is assigned to the hydrogen-linked urethane carbonyl group (C=O) [43,44], while the free carbonyl vibration modes are attributed to the IR band located at 1730 cm^−1^ [41,42,43,44,45]. Moreover, the IR band at 3300 cm^−1^ is associated with the NH group [45], which means that a partial inter- and intra-molecular hydrogen linkage of NH group is made between the adjacent urethane segments. The advantage of the inter-molecular hydrogen bonds between neighboring polyurethane groups consists of more stable and segregation-free TPU segments. Due to the interconnected effects between TPU and TPO, in the TPU/TPO blend (Figure 6c,d), the IR absorption band at 3300 cm^−1^ decreased completely in the TPU:TPO 1:2 blend, while the IR vibrational modes of the hydrogen bond–CO group showed a continuous decrease in its corresponding IR band located at 1701 cm^−1^ with an increase in the TPO concentration in the TPU:TPO blend. This means that the hydrogen bonds between the adjacent urethane groups are significantly suppressed by the surrounding polyolefins segments of TPO, which are likely similar to a wrapped membrane that isolates the CO and NH groups. According to Figure 6, as the mass ratio of the two polymers is changed in favor of TPU, a modification in the ratio between the absorbance of the IR bands from 1009 and 1076 cm^−1^ can be observed. With an increase in the BaTiO_3_ NP concentration in the TPU:TPO 1:1 blend, the following changes can be reported: (i) a down-shift of the IR bands from 800, 1018, and 1074 cm^−1^ (Figure 6e) to 795, 1013, and 1061 cm^−1^ (Figure 7c); (ii) a gradual increase of the ratio between the absorbance of the two IR bands situated in the spectral range 950–1100 cm^−1^ from 0.71 (Figure 6c) to 1.4 (Figure 7c); (iii) a progressive decrease of the ratio between the absorbance of the IR bands peaked at 1221 and 1257 cm^−1^ from 1.2 (Figure 6c) to 0.74 (Figure 7a), 0.63 (Figure 7b) and 0.22 (Figure 7c); (iv) a decrease in the absorbance of the IR bands situated in the spectral range 1400–1730 cm^−1^ (Figure 6c and Figure 7). Such a variation can also be observed in Figure 6, when the mass ratio between the two thermoplastic polymers changed. This variation can be explained by taking into account an exchange reaction between TPU and TPO, as shown in Scheme 1.

Returning at Figure 7, the above variations indicate a chemical interaction between TPU and BaTiO_3_ NPs, which can be described as an exchange reaction that takes place according to Scheme 2. 

Regardless of the BaTiO_3_ NP concentration in the TPU:TPO 2:1 blend, the increase in the absorbance of the IR band localized in the spectral range 750–900 cm^−1^ indicates the steric hindrance effects induced in the macromolecular chain of TPO as a consequence of the chemical transformations shown in Scheme 2.

### 3.3. Photoluminescence Properties of the Composites Based on the TPU:TPO Blends and BaTiO_3_ NPs

Figure 8 presents the PL spectra of TPU, TPO, and the BaTiO_3_ NPs recorded at the excitation wavelength equal to 350 nm. The PL spectra of TPU, TPO, and the BaTiO_3_ NPs showed a band with the maximum centered at 412 nm (Figure 8a), 441 nm (Figure 8b), and 530 nm (Figure 8c), respectively. The PL band of TPU at 412 nm (Figure 8a) was situated very close to the PL band of polyurethanes containing diphenylmethane 4,4′-diisocyante at 412 nm [46]. The origin of the PL band of TPU is probably due to the intrinsic chromophore centers based on benzene rings and carbonyl groups from the urethan segment of TPU, determining the appearance of a strong intensity of ~3.97 × 10^6^ counts/s. Contrary to expectation, TPO also showed a strong PL band, reaching a maximum intensity of ~2.38 × 10^6^ counts/s, even if the intrinsic chemical structure did not have luminous centers. However, one possible speculation of this result may be the material surface oxidation or the formation of extrinsic carbonyl groups at the TPO surface due to the recorded PL spectrum under normal conditions, namely in the presence of O_2_ and water vapor. A strong PL was also observed in polyethylene and propylene [47,48] and the explanation was reported taking into account the susceptibility of O_2_ in the air to the surface of these polymers.

In turn, the broad PL band of BaTiO_3_ in the visibly wide range has also been reported in the literature [49,50], and was attributed to the electron-hole recombination from the predominant TiO_6_ clusters observed in the ordered crystalline structure of BaTiO_3_ responsible for the weak PL band of the material [50]. From a careful analysis of Figure 8 and Figure 9, the PL band becomes wider for films based on the TPU: TPO blends than the TPU and TPO counterparts, most probably due to the intermembrane transitions and interconnected effects between TPU and TPO. Figure 9 shows the dependence of the PL spectra of the films based on the TPU:TPO blends with the TPU concentration in the mixture of the two thermoplastic polymers. The following variations are highlighted in Figure 9: (i) an enhancement in the intensity of PL spectra of the TPU:TPO 2:1, TPU:TPO 4:1, and TPU:TPO 6:1 blends of 1.25, 4, and 7 times higher than that of the TPU-TPO 1:1 and (ii) a down-shift of the maximum of the emission band from 486 nm to 462 nm, 433 nm and 425 nm, when the concentration of TPU increased in the mixtures of the two thermoplastic polymers labeled TPU:TPO 1:1, TPU:TPO 2:1, TPU:TPO 4:1, and TPU:TPO 6:1, respectively. The intensity of the PL spectrum of the TPU:TPO 2:1 blend with the maximum at 462 nm was equal to 3.01 × 10^7^ counts/s (red curve in Figure 9).

As the BaTiO_3_ NP concentration in the TPU:TPO 2:1 blend increased, the following variations can be observed in Figure 10: (i) a gradual down-shift of the PL band to 459, 445, 439, and 420 nm, when the concentration of BaTiO_3_ NPs in the TPU:TPO blend was equal to 6.25, 12, 25, and 30 wt.%, respectively; and (ii) a progressive decrease in the intensity of the PL band at 2.85 × 10^7^, 1.6 × 10^7^, 8.41 × 10^6^, and 5 × 10^6^ counts/s, when the concentration of BaTiO_3_ NPs in the TPU:TPO blend was equal to 6.25, 12, 25, and 30 wt.%, respectively.

These changes have their origin in the recrystallization process of BaTiO_3_ NPs in the matrix of thermoplastic polymers, when an uneven crystallization and aggregation of the BaTiO_3_ NPs in different places of the membrane matrix and/or the interfacial electron transfer efficiency between BaTiO_3_ and the TPU:TPO blends occurs.

A down-shift of the PL band of the film based on the TPU:TPO 4:1 blend and BaTiO_3_ NPs with the concentration of 12 wt.% was noted to take place from 433 nm (green curve in Figure 9) to 425 nm (Figure 11). This behavior was not observed in the case of the film based on the TPU:TPO 6:1 blend and the BaTiO_3_ NPs with the concentration of 12 wt.%. In both cases, as shown in Figure 11, the presence of BaTiO_3_ NPs induced a decrease in the intensity of the PL band of the TPU:TPO 4:1 and TPU:TPO 6:1 blends. This process is more intense when the TPU weight in the TPU:TPO blend increases. An explanation for this process must consider the intrinsic influence of the chromophore groups of TPU. Considering the intermolecular interaction in TPU, the π electrons of the phenyl rings in the adjacent urethane segments located above and below the TPU sheets tend to overlap with an increase in the TPU concentration in the composite films, thus favoring these non-covalent interactions. Moreover, the π–π interactions of phenyl groups are connected with the carbonyl units coming from the intrinsic chemical structure of TPU, being stabilized and isolated by the surrounding polyolefins segments similar to a coiled membrane. This explanation takes into account the PL generation of polymers [51]. High-mobility π electrons absorb more light and therefore emit a strong PL after relaxation of the photoexcited electrons. We suppose that this is the reason why the PL is suppressed at a low concentration of TPU in the TPU:TPO blends as well as in the films based on the TPU:TPO blends and BaTiO_3_ NPs.

### 3.4. Dielectric Properties of the Composites Based on the TPU:TPO Blends and BaTiO_3_ NPs

The studied samples show a great variation of the real and imaginary components of the complex dielectric permittivity, ε*=ε′−iε″, both depending on the frequency and concentration of the BaTiO_3_ NPs. The real part, ε′, is called the dielectric constant while the imaginary part, ε″, is called the dielectric loss. In order to observe the influence of the composition of the sample through the concentration of the BaTiO_3_ NPs, we chose an electrical quantity suitable for this purpose. Experimental data on electrical properties of the films containing the TPO:TPU blends and BaTiO_3_ NPs are suggested by means of dielectric loss spectra depending on the frequency: ε″=ε″(f), or ε″=ε″(ω), where ω=2πf. For both variables, the logarithmic scale is used.

The analysis of the dielectric permittivity spectra was done with the help of Havriliak–Negami fitting functions (H–N) [52,53] with two components [54,55] corresponding to the relaxation processes observed in graphic representations:(1)$$\varepsilon_{HN}(\omega )=\{\varepsilon_{\infty ,1}+\frac{\Delta \varepsilon_{1}}{{\left[1+{\left(i\omega \tau_{\max,1}\right)}^{\alpha_{1}}\right]}^{\beta_{1}}}\}+\{\varepsilon_{\infty ,2}+\frac{\Delta \varepsilon_{2}}{{\left[1+{\left(i\omega \tau_{\max,2}\right)}^{\alpha_{2}}\right]}^{\beta_{2}}}\}$$
where $\Delta \varepsilon =\varepsilon_{S}-\varepsilon_{\infty }$ is the dielectric strength; $\varepsilon_{S}=\underset{\omega \to 0}{\lim}\varepsilon \prime (\omega )$ and $\varepsilon_{\infty }=\underset{\omega \to \infty }{\lim}\varepsilon \prime (\omega )$; the characteristic time, $\tau_{\max}$, is given by the position, in frequency, of dielectric losses maximum $\varepsilon \prime \prime (\omega_{\max})=\varepsilon {\prime \prime }_{\max}$, $\varepsilon \prime \prime (\omega_{\max})=\varepsilon {\prime \prime }_{\max}$; and the exponents α_i_ and β_i_, called the shape parameters, influence the extension (widening) and the symmetry of the relaxation curve, around the maximum point of losses $0<\alpha_{i}\leq 1$, $0<\beta_{i}\leq 1$.

BaTiO_3_ is one of the most important ferroelectric materials with a high dielectric constant, high performance of piezoelectricity, and electro-optical effect [56]. High dielectric constant, behavior in frequency and temperature makes it a preferred material in the electronics manufacturing industry, suitable for the manufacture of ceramic miniaturized capacitor, frequency-adjustable devices, or as relaxors [57,58,59].

Figure 12 shows the experimental spectra of dielectric losses (black curves) and H–N fit functions (red curves) in the case of the films containing the TPU:TPO 2:1 blend in the absence and in the presence of the BaTiO_3_ NPs when the concentration of the inorganic compound was equal to 6.25, 12, 25, and 30 wt.%. Figure 12 indicates the existence of two maximum points in the dielectric loss spectra, which indicates the presence of two dielectric relaxation processes, one at low frequency below 3 Hz, and the other at high frequencies over 50 kHz. The first dielectric relaxation process was dominant and its maximum point changed depending on the BaTiO_3_ NP concentration. In the low frequency region, the H–N functions have a very good overlap on the experimental data, indicating a correct fit. The shape parameters’ value, exponents $\alpha_{1}$ and $\beta_{1}$, are close to the unit, indicating Debye-type dielectric relaxation processes.

In the order to observe the influence of the concentration of the BaTiO_3_ NPs on the electrical properties of the TPU:TPO 2:1 blends, we superimposed on the same graph the spectra of dielectric losses for all the studied samples in Figure 13. The spectrum of the TPO:TPU 2:1 blend showed a maximum point at a frequency of about 2 Hz (black curve). From Figure 13, it can be observed that the maximum point “moves” to a lower frequency when the concentration of BaTiO_3_ NPs increased. Characteristic time, $\tau_{\max}$, is an important parameter of the dielectric relaxation process. It has two properties that make it important to consider: (a) its value does not depend on the geometric dimension of the sample; in other words, the measurement error for the sample thickness or diameter does not influence it; and (b) the maximum point location in the dielectric loss spectra, with the help of fitting functions, is done with highest precision in comparison with other dielectric relaxation process parameters; the fitted value of relaxation time has the highest confidence.

Figure 14 shows the correspondence that exists between the change in the characteristic time value and the increase in the BaTiO_3_ NP concentration in the structure of the TPU:TPO 2:1 blend.

The experimental results allowed us to assume that the dielectric relaxation processes are dipolar in nature. We considered the electric dipoles of some segments of the polymer chain and the permanent electric dipoles of the BaTiO_3_ NPs. The shape of the dielectric permittivity spectra depends on the interaction between the two electrical moments of the TPU:TPO 2:1 host matrix and the BaTiO_3_ guest NPs. As the concentration of BaTiO_3_ NPs increases, the interaction energy with the polymer chain (segments) increases, which causes the host molecular mobility to decrease. The effect is to increase the characteristic time of the TPU:TPO 2:1 blend with the increase in BaTiO_3_ NP concentration.

### 3.5. Differential Scanning Calorimetry (DSC) Study of the Composites Based on the TPU:TPO Blends and BaTiO_3_ NPs

Figure 15 shows the DSC curves of the TPU:TPO 2:1 blend and their composites with BaTiO_3_ NPs.

According to Figure 15, the results of the DSC analysis showed that the incorporation of the BaTiO_3_ NPs into the TPU:TPO matrix influenced the thermal stability of the samples. It was observed that the sample with 12 wt.% BaTiO_3_ was the most stable, having the highest value of melting temperature (Tm = 330.4°C). The maxima of melting peaks corresponding to the samples with 25 wt.% BaTiO_3_ and 30 wt.% BaTiO_3_ were very close (Table 1), but lower than that of the sample with 12 wt.% BaTiO_3_.

On the other hand, at 6.25 wt.% BaTiO_3_, Tm = 306.5 °C, lower than the melting temperature of the TPU:TPO 2:1 blend. It can be seen that the thermal stability of the TPU:TPO 2:1 blend to which was added the BaTiO_3_ NPs did not increase proportionally to the amount of inorganic compound, but a critical amount of the BaTiO_3_ NPs can lead to an increase in the thermal stability of the macromolecular compounds, as observed in the case of the sample TPU:TPO + 12 wt.% BaTiO_3_. Regarding the melting enthalpies, the TPU:TPO 2:1 blend with 12 wt.% BaTiO_3_ had the lowest value (ΔH = 33.14 J/g) confirming the stability of the material.

## 4. Conclusions

In this work, we reported new results concerning the optical, structural, and dielectric properties of the films containing the TPU:TPO blends and BaTiO_3_ NPs. Using SEM, Raman scattering, FTIR spectroscopy, photoluminescence, XRD, and dielectric analysis, the following conclusions can be drawn: (i) the SEM images highlighted that the free films of TPU and TPO are characterized by discrete phases of spherical shape, which are randomly distributed to form the cross-physical contacts between the amorphous segments of TPU and TPO; (ii) XRD analysis demonstrated that the BaTiO_3_ NPs had a tetragonal structure that was not disturbed when they were dispersed in the films based on the TPU:TPO blends; (iii) according to the Raman scattering and FTIR spectroscopy studies, an exchange reaction was invoked to take place during the preparation of the films based on the TPU:TPO blends; in the presence of the BaTiO_3_ NPs, this exchange reaction induces the appearance in the addition of the compound BaTiO_3-m_; (iv) an enhancement in the PL bands intensities of the TPU:TPO 2:1, TPU:TPO 4:1, and TPU:TPO 6:1 blends of 1.25, 4, and 7 times higher than that of the TPU-TPO 1:1 accompanied by a down-shift of the maximum of the emission band from 486 nm to 462 nm, 433 nm and 425 nm, when the TPU concentration increased in the two thermoplastic polymer blends labeled TPU:TPO 1:1, TPU:TPO 2:1, TPU:TPO 4:1, and TPU:TPO 6:1, respectively, were reported; our studies indicate that the BaTiO_3_ NPs play the role of the PL quenching agent of the TPU:TPO blends; and (v) the host matrix of the TPO:TPU 2:1 blend and their composites with the BaTiO_3_ NPs showed two distinct processes of well separated dipolar relaxation: one at low frequencies and another at high frequencies. For low frequency relaxation process, the characteristic time increased with an increase in the BaTiO_3_ concentration.

## Data Availability

The data reported in this article are available on request from the corresponding author.

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
