# Peer review of "Optical, Structural, and Dielectric Properties of Composites Based on Thermoplastic Polymers of the Polyolefin and Polyurethane Type and BaTiO3 Nanoparticles"

_materials, 2021, doi:10.3390/ma14040753_

Round 1

Reviewer 1 Report

This is a detailed work about TPU/TPO and BaTiO3 nanoparticles with characterization on the optical, structural and dielectric properties. The reviewer has the following questions for the authors to respond: 1) Page 2, line 67, what is the size of the BaTiO3 nanoparticles? Have the authors estimated that from SEM and XRD? 2) Page 3, line 115: the use of “thermodynamic incompatibilities” is poorly defined, can the author more specifically explain what is that. 3) Page 3, line 122: The SEM figures, particularly (b) ~ (d) are very blurry, can the authors state why it was the case, or redo the SEM. Where the BaTiO3 are distributed in the TPU:TPO? The “uniform dispersion of BaTiO3” does not seem to be obvious, perhaps a low-magnification SEM can help reveal the dispersion pattern. 4) Page 3, SEM figure, it would help readers if the authors can identify phases in the SEM images (e.g. by using arrows and texts) 5) Page 5, line 155, in figure 2 (b), what is the peak near 30 degree? Such peak, however, does not show up in blend TPU:TPO, nor the one mixed with BaTiO3. 6) Page 5, line 155, in figure 2(e), why there are double peaks around 40 degree? Note that from the pure BaTiO3 XRD, around 40 degree, there is only (111) diffraction. Editorial comments: The authors should also check typos and errors throughout the manuscript.

Author Response

Comments and Suggestions for Authors

This is a detailed work about TPU/TPO and BaTiO3 nanoparticles with characterization on the optical, structural and dielectric properties. The reviewer has the following questions for the authors to respond:

1) Page 2, line 67, what is the size of the BaTiO3 nanoparticles? Have the authors estimated that from SEM and XRD?

Authors reply: The size of BaTiO3 nanoparticles have been determined from HRTEM. In the revised manuscript, we have included the following comments:

- lines 1327-137:

“BaTiO3 NPs were studied by high-resolution transmission electron microscopy (HRTEM). In this order a suspending of BaTiO3 NPs in C2H5OH was prepared and successively this was transferred onto a copper grid coated with amorphous carbon support. HRTEM images BaTiO3 NPs were recorded with a JEOL JEM ARM 200 F electron microscope (JEOL (Europe) SAS), which works at 200 keV. “

-lines 217-223:

“In order to shown additional information concerning the inorganic NPs size, in Figure 2 is shown the HRTEM image of the BaTiO3 NPs. The analysis of Figure 2 indicates in the case of the number of count particles equal to 100, the values of the minimum, maximum and mean size of BaTiO3 NPs equal to 30.04 ± 9.89 nm, 79.93 ± 9.89 nm and 53.35 ± 9.89 nm, respectively.

Figure 2. HRTEM images of the BaTiO3 NPs. “

2) Page 3, line 115: the use of “thermodynamic incompatibilities” is poorly defined, can the author more specifically explain what is that.

Authors reply: Line 115 is localized at line 188 in the revised manuscript. The sentence form line 185 has been rewritten as follows at lines 188-197:

“Taking into account the TPU composition, it is composed of hard polar segments (urethane component) and soft less-polar segments (long-chain diol) with glassy and rubbery character, respectively. Such a structure is described by a high and low glass transition temperature that induces the thermodynamic incompatibility between hard and soft segments. In this context, self-assembly of hard segments through physical bonds causes the Gibbs free energy of the film to become positive, inducing a two-phase separation with the formation of a hard-micro-domains aggregation of the urethane segments [28]. “

3) Page 3, line 122: The SEM figures, particularly (b) ~ (d) are very blurry, can the authors state why it was the case, or redo the SEM. Where the BaTiO3 are distributed in the TPU:TPO? The “uniform dispersion of BaTiO3” does not seem to be obvious, perhaps a low-magnification SEM can help reveal the dispersion pattern.

Authors reply: In the revised manuscript, we have included new SEM images for the films of TPU:TPO containing a concentration of BaTiO3 nanoparticles equal to 6 wt.%, 12 wt.%, 25 wt.% and 30 wt.%. We have indicated by arrows the region where there is only TPU:TPO and regions where exist the BaTiO3 nanoparticles in the TPU:TPO matrix. Please, see all these changes in Figure 1.

4) Page 3, SEM figure, it would help readers if the authors can identify phases in the SEM images (e.g. by using arrows and texts)

Authors reply: In the revised manuscript, we have indicated by arrows and texts the phases of BaTiO3 nanoparticles in Figure 1.

5) Page 5, line 155, in figure 2 (b), what is the peak near 30 degree? Such peak, however, does not show up in blend TPU:TPO, nor the one mixed with BaTiO3.

Authors reply: In the revised manuscript, we have included the following comment at lines 233-237:

“In addition, Figure 3b shows a series of small maxima, corresponding most probably, to a minor orderd phase formed inside the TPU polymer. The most proeminent maximum from this series is the first one, at around 29.3°, followed by smaller ones at 35.9°, 39.3°, 43.1° and 47.3°. These data correlate rather well with a TPU diffraction pattern reported in [29].  “

6) Page 5, line 155, in figure 2(e), why there are double peaks around 40 degree? Note that from the pure BaTiO3 XRD, around 40 degree, there is only (111) diffraction.

Authors reply: In the revised manuscript, in Figure 2e has been marked with * the peaks at 38.3° and 44.5°, which belongs to the sample holder of the diffractometer. At line 253, we have included the following comment

“The peaks,  marked with *, at 38.3° and 44.5° belongs to the sample holder of the diffractometer.“

Editorial comments: The authors should also check typos and errors throughout the manuscript.

Authors reply: In the revised manuscript, all check typos and errors throughout the manuscript have been corrected. These are marked with track changes in the manuscript. Please, see lines 23, 25- 27, 30-34, 36, 43-48, 51, 52, 57, 61- 64, 67-81, 83-89, 92, 96- 105, 110-111, 113- 116, 119-120, 122-127, 130-162, 172-177, 181, 182, 189-195, 202, 213-215, 217-223, 228, 231-236, 252-265, 277, 284-285, 293, 295, 330-349, 365, 369-382, 402-413, 426- 432, 438, 439, 443-451, 464-475, 494, 500, 504-505, 508, 513, 525-529, 532-566, 570, 573, 576, 577, 579, 580, 587 and 589.

Reviewer 2 Report

The authors report on the optical, structural and dielectric properties of the composites based on thermoplastic polymers, polyurethane and BaTiO3 nanoparticles.

The work is quite innovative for the field of functional polymer nanocomposites, while there is quite a lot of experiments behind this study.

Some suggestions below that should be included/ revised:

  1. It is not correct at all the first sentence of the manuscript – lines 38-39. First of all, the technique scientifically is called “fused filament fabrication” and not FDM (this is a commercial name/ definition used by a specific 3D printer company supplier). Moreover, TPU is not interesting only for 3D FFF printing applications. Moreover, what is the meaning of “3D inkjet printing devices”; 3D FFF printing is a printing technique based on an extruder, so it can be defined in other words as “melt printing technique” but not ink. Why also there are needed [1-4] references the authors to support this first sentence of the paper – introduction Part. Please be careful throughout the whole manuscript with the use of proper and references that you use.
  2. Line 41: it is not “interaction” – it is the product of the polymerisation reaction of ... Once more, please be careful with your statements especially in the introduction part, it is very important that you write correct “scientific” things together with the appropriate literature since this part is educative for the audience and such mistakes may mislead the community to wrong information/ knowledge.
  3. Line 44: “The heat resistance as well as the stiffness of TPU are lower”, please use formal language; i.e. “The heat resistance as well as the stiffness of TPU are relatively low [ref]”... and please it is not a “mixture” with other polymers but blending.
  4. Line 50: “low weight”.. maybe low density or “high specific properties” and then mention some of them that you are also focusing in the study at hand.
  5. Line 54: please replace “sustained” with e.g. “continuous effort has been devoted”
  6. Line 57: “for applications in the field of 3D printing devices,” what is the literature there... possibly please consider some recent literature (Materials & Design, Volume 131, 5, 2017, Pages 394-401, Materials 2020, 13(12), 2879) for 3D FFF printed TPU nanocomposite materials.
  7. Line 59: please put the full stop after the reference as in the whole document.
  8. Please for the term “nanoparticles”, abbreviate it one in the abstract or in introduction as “NPs” and then use it in its abbreviated form throughout the whole manuscript.
  9. Line 59-64: the last two paragraphs of introduction, please use more formal language.
  10. Please the whole introduction part should be revised with better language use to improve grammatical errors, some words replacement and more formal English. Moreover, sentences should be re-written and more literature is needed to highlight the state of the art literature for this type of nanocomposites. As such, maybe two or three additional paragraphs are needed to extend/ expand the introduction part that should be educative to the audience. For instance, mention techniques for fabricating the BaTiO3 nanocomposites i.e. solvent mixing or melt mixing; some physical property values, etc.
  11. For Raman and FT-IR – 3.2 section is really long, please plot in one graph the different materials you compare, it is very difficult for the reader to follow.
  12. The SEM images are really of bad quality, please try to improve them.
  13. Line 270-273, please include the appropriate literature for the PL of TPU.
  14. Throughout the whole manuscript, please first give you interpretation – discussion of the results and then the respective figure. This is not the case for Figure 8.
  15. Throughout the whole manuscript, I propose to use the term “film” and not membrane. Moreover how these films have been produced and what is their thickness. Does the thickness play a role?
  16. Is it possible the authors to include some TEM images to demonstrate the “nanodispersion” state and the quality of “nanodispersion” of the BaTiO3 nanoparticles?
  17. For nanocomposite samples, DSC is also typically performed to give very important information on the thermoplastic matrix, how it is affected by the presence of “nanofiller”, etc. It will be nice the authors to include some DSC analysis.

In terms of originality, importance & scientific quality, relevance & contribution to the field and presentation, this manuscript is of good level.

However, the quality of the manuscript in its current form is not the proper one to be published in such a high impact factor journal.

The manuscript should be improved in the points that have been indicated in order to make it more interesting to the reader and more educative, while improving the quality overall.

The manuscript and its content are sufficiently novel to warrant its publication, however, after including and considering the additions and clarifications proposed.

Author Response

Review 2

The authors report on the optical, structural and dielectric properties of the composites based on thermoplastic polymers, polyurethane and BaTiO3 nanoparticles.

The work is quite innovative for the field of functional polymer nanocomposites, while there is quite a lot of experiments behind this study.

Some suggestions below that should be included/ revised:

    It is not correct at all the first sentence of the manuscript – lines 38-39. First of all, the technique scientifically is called “fused filament fabrication” and not FDM (this is a commercial name/ definition used by a specific 3D printer company supplier). Moreover, TPU is not interesting only for 3D FFF printing applications. Moreover, what is the meaning of “3D inkjet printing devices”; 3D FFF printing is a printing technique based on an extruder, so it can be defined in other words as “melt printing technique” but not ink. Why also there are needed [1-4] references the authors to support this first sentence of the paper – introduction Part. Please be careful throughout the whole manuscript with the use of proper and references that you use.

Authors reply: In the revised manuscript, the sentence from lines 38-39

“The main interest for thermoplastic polyurethane (TPU) and their nanocomposites was in the fused filament deposition in the 3D inkjet printing devices. [1-4]”

has been rewritten as follows at the lines 45-47

“The main interest for thermoplastic polyurethane (TPU) and their nanocomposites was in the field of fused filament fabrication, electronic devices, the automotive panels, sporting goods, belts, profile, tubes, hoses and so on. [2- 4] “  

    Line 41: it is not “interaction” – it is the product of the polymerisation reaction of ... Once more, please be careful with your statements especially in the introduction part, it is very important that you write correct “scientific” things together with the appropriate literature since this part is educative for the audience and such mistakes may mislead the community to wrong information/ knowledge.

Authors reply: In the revised manuscript, the sentence at line 41

“TPU is a linear elastomer with both the non-polar soft and polar hard segments, resulted by the interaction of diisocyanates with the diols with short and long-chains, respectively.” has been rewritten at line 42-45 as follows:

„TPU is a linear elastomer with both the non-polar soft and polar hard segments, resulted by the polymerization reaction  of diisocyanates with the diols with short and long-chains, respectively [1, 2].”

    Line 44: “The heat resistance as well as the stiffness of TPU are lower”, please use formal language; i.e. “The heat resistance as well as the stiffness of TPU are relatively low [ref]”... and please it is not a “mixture” with other polymers but blending.

Authors reply: Line 44 is localized at line 50 in the revised manuscript. The sentence has been rewritten as follows:

“The heat resistance as well as the stiffness of TPU are relative low and therefore the blend with other polymer matrix is seen as a way attractive to improve these properties [8, 9]. “

    Line 50: “low weight”.. maybe low density or “high specific properties” and then mention some of them that you are also focusing in the study at hand.

Authors reply: Line 50 in the revised manuscript is localized at line 55. The sentence has been rewritten as follows:

„Thermoplastic polyolefin (TPO), known as polyethylene-polyoctene rubbers, is one of the most common and commercial thermoplastic polymer used due to its high rigidity and good chemical resistance [15, 16].”

    Line 54: please replace “sustained” with e.g. “continuous effort has been devoted”

Authors reply: Line 54 is localized in the revised manuscript on line 61. This line has been rewritten as follows:

“Since 2013, a continuous effort has been devotedin the development of new composites based on the BaTiO3 nanoparticles (NPs), thermoplastic polymers of the type polyvinylidene fluoride (PVDF) and various carbon NPs such as graphene [20] or carbon nanotubes [21] for applications in the field of 3D printing devices [3, 22], medical devices [21] and textile fibers [21].“

    Line 57: “for applications in the field of 3D printing devices,” what is the literature there... possibly please consider some recent literature (Materials & Design, Volume 131, 5, 2017, Pages 394-401, Materials 2020, 13(12), 2879) for 3D FFF printed TPU nanocomposite materials.

Authors reply: Line 57 in the revised manuscript is localized at line 61. This line has been rewritten as follows:

“Since 2013, a continuous effort has been devotedin the development of new composites based on the BaTiO3 nanoparticles (NPs), thermoplastic polymers of the type polyvinylidene fluoride (PVDF) and various carbon NPs such as graphene [20] or carbon nanotubes [21] for applications in the field of 3D printing devices [3, 22], medical devices [21] and textile fibers [21].“

[22] Christ, J. F.; Aliheidari, N.; Ameli, A.; Pötschke, P. 3D printed highly elastic strain sensors of multiwalled carbon nanotube/thermoplastic polyurethane nanocomposites. Mater. Design 2017, 131, 394–401. http://dx.doi.org/10.1016/j.matdes.2017.06.011.

In the initial manuscript, the reference Materials 2020, 13(12), 2879 was noted as Reference [3].

    Line 59: please put the full stop after the reference as in the whole document.

    Please for the term “nanoparticles”, abbreviate it one in the abstract or in introduction as “NPs” and then use it in its abbreviated form throughout the whole manuscript.

Authors reply: In the revised manuscript, we have included a new figure, which shows the BaTiO3 nanoparticles shape. The following comment has been included at the lines 23, 27, 58, 59, 68, 71, 82, 94, 100, 106, 110, 142, 159, 161, 175, 186, 187, 209, 211, 215, 217, 220, 251, 286, 303, 319, 322, 329, 330, 350, 379, 382, 385, 387, 391, 392, 396, 400, 402, 404, 418, 424, 427, 429, 448, 453, 459, 461, 466, 477, 481, 485, 487, 490, 492, 494, 498, 502, 508, 512, 514, 522, 527, 531, 538, 540.

    Line 59-64: the last two paragraphs of introduction, please use more formal language.

    Please the whole introduction part should be revised with better language use to improve grammatical errors, some words replacement and more formal English. Moreover, sentences should be re-written and more literature is needed to highlight the state of the art literature for this type of nanocomposites. As such, maybe two or three additional paragraphs are needed to extend/ expand the introduction part that should be educative to the audience. For instance, mention techniques for fabricating the BaTiO3 nanocomposites i.e. solvent mixing or melt mixing; some physical property values, etc.

Authors reply: In the revised manuscript, the following comments have been included at lines 68-88:

“As known well, BaTiO3 is characterized by a high permittivity of about 105 and a low dielectric loss of ~0.05 [23]. Due to the low polarization ability and high dielectric loss of thermoplastic elastomers, the incorporation of highly polarizable filler like BaTiO3 NPs in the thermoplastic polymers matrix promotes interfacial (exchange coupling) effects that could significantly enhance the final dielectric performance of the resulting blends for specific electro-active applications [20].

The most widely used solvent mixing techniques for BaTiO3 nanocomposites are based on the solution casting method or in situ polymerization [20]. At the industrial level, the improvement of the manufacturing performance of polymer blends has led to a melt mixing technique using co-rotating twin screw extruder for the fabrication of BaTiO3 - elastomers nanocomposites [24]. In this context, we note that it has already found a high real dielectric permittivity (ɛr=5.06) and a low dielectric loss (less than 0.05) in the case BaTiO3 incorporated poly(styrene-ethylene/butylene-styrene)-grafted-maleic anhydride elastomers [24] and a significant dielectric permittivity at low frequency (values exceeding 104) in the PVDF/ BaTiO3 composites [20].

The realization of applications presupposes a good knowledge of the crystallization process of BaTiO3 in the polyvinylidene fluoride matrix. [25] In the case of the PVDF/BaTiO3 composites, the adding inorganic particles to the matrix of the macromolecular compound has been demonstrated to induce a change in the process of interfacial polarization and in the crystallization kinetic of heterogenous nucleation, when a wider range of melting temperatures of the polymer in the presence of BaTiO3 NPs was reported. [20, 25] Knowledge of the crystallization process of the BaTiO3 NPs in the thermoplastic polymers matrix of the type TPU and TPO can open perspectives for new applications as that of 3D and 4D printed devices. [4, 26] Therefore, preliminary studies focused on the optical, structural and dielectric properties of composites based on the TPU:TPO blends and the BaTiO3 NPs will be shown in this work.“

    For Raman and FT-IR – 3.2 section is really long, please plot in one graph the different materials you compare, it is very difficult for the reader to follow.

Authors reply: Figures 6 has been changed according to above comment.

    The SEM images are really of bad quality, please try to improve them.

Authors reply: In the revised manuscript, new SEM images for the samples of TPO : TPU 2:1 containing 6.25 wt.% (Figure 1b), 12 wt.% (Figure 1c) and 25 wt.% BaTiO3 (Figure 1d) nanoparticles have been introduced in Figure 1.

    Line 270-273, please include the appropriate literature for the PL of TPU.

Authors reply: Lines 270-273 in the revised manuscript are situated at line 376-378. In the revised manuscript we have included the following comment:

“The PL bandof TPU at 412 nm (Figure 8a) is situated very close to the PL band of polyrethanes containing diphenylmethane 4, 4’-diisocyante at 412 nm [46]. “

[46] Allen N.S.; McKellar, J.F. Photochemical reactions in an MDI-based elastomeric polyurethane, J. Appl. Polym. Sci. 1976, 20, 1441-1447.

    Throughout the whole manuscript, please first give you interpretation – discussion of the results and then the respective figure. This is not the case for Figure 8.

Authors reply: Figure 8 corresponds to Figure 9 in the revised manuscript. Figure 9 has been situated after interpretation as observed in lines 400-409:.

“Figure 9 shows the dependence of the PL spectra of the films based on the TPU:TPO blends with the TPU concentration in the mixture of the two thermoplastic polymers. The following variations are highlighted in Figure 9: i) an enhanced in the intensity of PL spectra of the blends TPU:TPO 2:1, TPU:TPO 4:1 and TPU:TPO 6:1 of 1.25, 4 and 7 times higher than that of the TPU-TPO 1:1 and ii) a down-shift of the maximum of the emission band from 486 nm to 462 nm, 433 nm and 425 nm, when the concentration of TPU increases in the mixtures of the two thermoplastic polymers labeled TPU:TPO 1:1, TPU:TPO 2:1, TPU:TPO 4:1 and TPU:TPO 6:1, respectively. The intensity of the PL spectrum of the TPU:TPO 2:1 blend with the maximum at 462 nm is equal to 3.01 x 107 counts/sec (red curve in Figure 9).”

    Throughout the whole manuscript, I propose to use the term “film” and not membrane. Moreover how these films have been produced and what is their thickness. Does the thickness play a role?

Authors reply: In the revised manuscript, the word “membrane” has been replaced with “film”. Please, see lines 80, 85, 88, 94, 151, 156, 176, 212, 224, 231, 240, 293, 344, 385, 406, 417, 436, 446, 511, 514 and 518.

 Information about the preparation of films of TPU:TPO in absence and in the presence of BaTiO3 nanoparticles are shown in lines 104-124. Please, see below comment.

“2.2 The preparation of the films of TPU:TPO in absence and in the presence of BaTiO3 NPs

The TPU:TPO blends were prepared as free membranes as follows. In the first stage, it was prepared under ultrasonication, a solution of TPU in DMF (0.5 g/8 mL) and another of TPO in C2H5OH (0.5 g/10 mL), to which was added 0.1 g of the TPO hardener in 2 mL DMF. After mixing them ultrasonically for 5 min, the resulting solution was placed in a Petri dish and dried for 2 hours at 100 °C. The resulting film peeled off the Petri glass and labeled as the TPU:TPO 1:1 blend, this being dried under vacuum until to constant mass. Using this protocol and the changing  the mass ratio of the two thermoplastic polymers, five other free films labeled as the blends TPU:TPO 1:3, TPU:TPO 1:2, TPU:TO 2:1, TPU:TPO 4:1, TPU:TPO 6:1 were prepared. The thickness of all blends was of cca. 43 µm.

The composites based on the TPU:TPO 1:1 blends and BaTiO3 NPs were prepared as above described, the only difference being that after the mixture of the TPU and TPO solutions, the various weights of inorganic particles, i.e. 0.05, 0.1, 0.23 and 0.3 g, were added under ultrasonication for 10 min. After the thermal treatment at 100 °C and the drying under vacuum until to constant mass, the film labeled as composites based on TPU:TPO 1:1 and BaTiO3, having the concentration of inorganic NPs equal to 6.25, 12, 25 and 30 wt.%, respectively, were obtained. As increasing the BaTiO3 NPs concentration in the polymers mass, from 6.25, 12, 25 and 30 wt.%, the thickness of films was changed at cca. 43.4, 44, 44.6 and 45 µm. “

    Is it possible the authors to include some TEM images to demonstrate the “nanodispersion” state and the quality of “nanodispersion” of the BaTiO3 nanoparticles?

Authors reply: In the revised manuscript has been included also a new figure with the TEM images of the BaTiO3 nanoparticles (Figure 2) and the following comments:

  • lines 132-137:

“BaTiO3 NPs were studied by high-resolution transmission electron microscopy (HRTEM). In this order a suspending of BaTiO3 NPs in C2H5OH was prepared and successively this was transferred onto a copper grid coated with amorphous carbon support. HRTEM images BaTiO3 NPs were recorded with a JEOL JEM ARM 200 F electron microscope (JEOL (Europe) SAS), which works at 200 keV.“

-lines 217- 223:

“In order to shown additional information concerning the inorganic NPs size, in Figure 2 is shown the HRTEM image of the BaTiO3 NPs. The analysis of Figure 2 indicates in the case of the number of count particles equal to 100, the values of the minimum, maximum and mean size of BaTiO3 NPs equal to 30.04 ± 9.89 nm, 79.93 ± 9.89 nm and 53.35 ± 9.89 nm, respectively.

    For nanocomposite samples, DSC is also typically performed to give very important information on the thermoplastic matrix, how it is affected by the presence of “nanofiller”, etc. It will be nice the authors to include some DSC analysis.

Authors reply: According to above request, the DSC studies have been included in the revised manuscript. Results are shown in Figure 14 and Table 1. Please, see the following comments in the revised manuscript:

- lines 172-178

2.9 Differential scanning calorimetry (DSC) investigations

DSC measurements were performed with a DSC 204 F1 type from Netzsch (Selb, Germany).The experiments were conducted from room temperature to 500 °C in an inert atmosphere of helium (40 mil/min flow gase rate) and a heating rate of 5°C/min.  The samples were sealed in Al crucibles by pressing and as reference was used an empty Al crucible. The accuracy of heat flow measurements was ±0.001 mW.“

-lines 536-563

“3.5. DSC study of the composites based on the TPU:TPO blends and BaTiO3 NPs

Figure 15 shows the DSC curves of the TPU:TPO 2:1 blend and their composites with BaTiO3 NPs.

 .....

Figure 15. DSC curves of the TPU:TPO 2:1 blend (black curve) and their composites with BaTiO3 NPs, the inorganic compound concentration being equal to 6.25 wt.% (red curve), 12 wt.% (blue curve), 25 wt.% (green curve) and 30 wt.% (magenta curve).

According to Figure 15, results of DSC analysis showed that the incorporation of the BaTiO3 NPs into the TPU:TPO matrix influenced the thermal stability of the samples. It was observed that the sample with 12 wt.% BaTiO3 is the most stable having the highest value of melting temperature (Tm=330.4°C). The maximum of melting peaks corresponding to the samples with 25 wt.%BaTiO3 and 30 wt.% BaTiO3 are very close (Table 1) but lowers than the sample with 12 wt.% BaTiO3.

Table 1. The melting parameters estimated from DSC endothermic curves of the TPU:TPO 2:1 blend and their composites with BaTiO3 NPs

.......

On the other hand, at 6.25 wt.% BaTiO3, Tm = 306.5 °C, lower than the melting temperature of the TPU:TPO 2:1 sample. It can be seen that the thermal stability of the TPU:TPO 2:1 blend at which has been added the BaTiO3 NPs does not increase proportional to the amount of inorganic compound, but is a critical amount of the BaTiO3 NPs that can lead to an increase in the thermal stability of the macromolecular compounds, as observed in the case of the sample TPU:TPO + 12 wt.% BaTiO3. Regarding to the melting enthalpies, the sample with 12 wt.% BaTiO3 has the lowest value (ΔH=33.14 J/g) confirming the stability of the material. “

Reviewer 3 Report

The authors present a preliminary study regarding a composite of thermoplastic polymers and BaTiO3.

Major Comments:

The text needs to be revised regarding grammar and vocabulary.

Abstract: I would suggest the authors replace the word highlighted. It doesn’t express the idea they are seeking and it doesn’t sound scientific. Consider replacing it for: demonstrated, studied, showcased. Please keep consistency on how to represent BaTiO3. In some appearances, it is named BaTiO3

Introduction. I would suggest the authors change the order of the sentences. Start defining TPU and then cover their main application, not the opposite as it is. Also, by the end, it is not clear why inorganic nanoparticles are used. The authors could add a sentence just to help to understand why the inorganic materials are also added. There is a good explanation for the polymer mixture, but not as good for the addition of the nanoparticles to the composite.

Materials and methods: It needs to be added more details on manufacturers. Like: ......purchased by Company (City, Country)

I would recommend the authors to create sections for each method and each analytical measurement. Also, some analysis needs more details as SEM, IR, Raman. It should be included some sample preparation and machines setups

Results: The results are very well described and detailed. 

The schematic of the polymers could be improved. It is very hard to see what is described in the text. Consider making a scheme with different colors or different thicknesses of bonds. 

I believe that the polymer schematic could be presented at the beginning of the discussion, for clarity. Also, a schematic showing the film and inorganic particles could make it easier to understand the systems synthsized. 

Minor Comments:

Line 50. low weight, high rigidity or good chemical resistance. The use of “or” makes the sentence a little bit confusing. If appropriate consider change it to “and”

Line 53. Since the authors decide to use the date (2013), this affirmation needs a reference.

Line 56. Typo: graphene

Line 57. The word “health” needs more description. Are the authors intending to say medical devices?

Line 66. The hardener should be described (which chemical compound it is)

Line 71. Replace the chemical formula for ethanol.

Line 80. Please keep consistence on the figures, for clarity: 0.05, 0.10, 0.23, 0.30

Line 81. The symbol for the Celsius degree is wrong. Please replace the symbol (it is not a superscribed  zero or “o”)

Line 140. Use appropriate degree symbol, not superscribed Zeros or O

Line 160. Please keep the text impersonal. As expected

Author Response

Reviewer 3

The authors present a preliminary study regarding a composite of thermoplastic polymers and BaTiO3.

 Major Comments:

The text needs to be revised regarding grammar and vocabulary.

Authors reply: In the revised manuscript, a check of the English language has been performed. These are marked with the track changes. Please, see lines 55, 59, 62, 63, 80-85, 95-96, 100, 106, 109, 115, 116, 123-124, 159, 176-177, 182, 208, 209, 210, 223, 240, 242, 244, 249, 252, 253, 265, 272, 273, 281, 283, 351, 361, 362, 387, 411, 414, 417, 419, 423, 424, 428, 432, 434, 436, 449, 450, 453, 456, 459, 460, 461, 479, 485, 490, 494, 498, 507, 509, 513, 516, 518-519, 522, 545, 547, 555, 557, 560, 561, 563, 564, 571, 573.

Abstract: I would suggest the authors replace the word highlighted. It doesn’t express the idea they are seeking and it doesn’t sound scientific. Consider replacing it for: demonstrated, studied, showcased. Please keep consistency on how to represent BaTiO3. In some appearances, it is named BaTiO3

Authors reply: In the revised manuscript, according above comment, we have changed as follows:

a) the lines 23-25

“The vibrational properties of the free films as depending on the weight ratio of the two thermoplastic polymers were studied.”

b) lines 28-30

“The vibrational changes induced to the thermoplastic polymer’s matrix of the BaTiO3 NPs were showcased by Raman scattering and FTIR spectroscopy.”

c) lines 33-34

“The dependences of the dielectric relaxation phenomena with the BaTiO3 NPs weight in the thermoplastic polymers blends were also demonstrated.”

Introduction. I would suggest the authors change the order of the sentences. Start defining TPU and then cover their main application, not the opposite as it is. Also, by the end, it is not clear why inorganic nanoparticles are used. The authors could add a sentence just to help to understand why the inorganic materials are also added. There is a good explanation for the polymer mixture, but not as good for the addition of the nanoparticles to the composite.

Authors reply: In the revised manuscript, the order of the sentences from lines 38-41 of the initial manuscript 7 have been changed at line 42-47 as follows:

“TPU is a linear elastomer with both the non-polar soft and polar hard segments, resulted by the polymerization reaction  of diisocyanates with the diols with short and long-chains, respectively [1, 2]. The main interest for thermoplastic polyurethane (TPU)  and their nanocomposites was in the field of fused filament fabrication, electronic devices, the automotive panels, sporting goods, belts, profile, tubes, hoses and so on. [2- 4] “

The following sentence which to help to understand why the inorganic materials are added at the thermoplastic polymers has been included at lines 82-89, as follows:

“The realization of applications presupposes a good knowledge of the crystallization process of BaTiO3 in the polyvinylidene fluoride matrix. [25] In the case of the PVDF/BaTiO3 composites, the adding inorganic particles to the matrix of the macromolecular compound has been demonstrated to induce a change in the process of interfacial polarization and in the crystallization kinetic of heterogenous nucleation, when a wider range of melting temperatures of the polymer in the presence of BaTiO3 NPs was reported. [20, 25] “

Materials and methods: It needs to be added more details on manufacturers. Like: ......purchased by Company (City, Country)

I would recommend the authors to create sections for each method and each analytical measurement. Also, some analysis needs more details as SEM, IR, Raman. It should be included some sample preparation and machines setups.

Authors reply: In the revised manuscript, the section of Materials and Methods has been rewritten as follows

2. Materials and Methods

2.1. Materials

The thermoplastic elastomers, i.e. TPU and TPO (including its hardner which corresponds to polydimethylsiloxane, marketed under the name of Sylgard TM 186 Silicone elastomer curring agent), were purchased from Elastollan-BASF Chemical Company (Cleveland, OH, SUA). The BaTiO3 nanopowder, N,N'-dimethyl formamide (DMF) and C2H5OH were purchased from Sigma Aldrich (St. Louis, MO, USA ).

2.2 The preparation of the films of TPU:TPO in absence and in the presence of BaTiO3 NPs

The TPU:TPO blends were prepared as free membranes as follows. In the first stage, it was prepared under ultrasonication, a solution of TPU in DMF (0.5 g/8 mL) and another of TPO in C2H5OH (0.5 g/10 mL), to which was added 0.1 g of the TPO hardener in 2 mL DMF. After mixing them ultrasonically for 5 min, the resulting solution was placed in a Petri dish and dried for 2 hours at 100 °C. The resulting film peeled off the Petri glass and labeled as the TPU:TPO 1:1 blend, this being dried under vacuum until to constant mass. Using this protocol and the changing  the mass ratio of the two thermoplastic polymers, five other free films labeled as the blends TPU:TPO 1:3, TPU:TPO 1:2, TPU:TO 2:1, TPU:TPO 4:1, TPU:TPO 6:1 were prepared. The thickness of all blends was of cca. 43 µm.

The composites based on the TPU:TPO 1:1 blends and BaTiO3 NPs were prepared as above described, the only difference being that after the mixture of the TPU and TPO solutions, the various weights of inorganic particles, i.e. 0.05, 0.1, 0.23 and 0.3 g, were added under ultrasonication for 10 min. After the thermal treatment at 100 °C and the drying under vacuum until to constant mass, the film labeled as composites based on TPU:TPO 1:1 and BaTiO3, having the concentration of inorganic NPs equal to 6.25, 12, 25 and 30 wt.%, respectively, were obtained. As increasing the BaTiO3 NPs concentration in the polymers mass, from 6.25, 12, 25 and 30 wt.%, the thickness of films was changed at cca. 43.4, 44, 44.6 and 45 µm.

2.3 Microscopy investigations

The scanning electron microscopy (SEM) images of the BaTiO3 NPs and their composites with the PTU:TPO blends were recorded with a scanning electron microscope, model Tescan Lyra III XMU (Libušina tĹ™. 21 623 00 Brno - Kohoutovice, Czech Republic).

BaTiO3 NPs were studied by high-resolution transmission electron microscopy (HRTEM). In this order a suspending of BaTiO3 NPs in C2H5OH was prepared and successively this was transferred onto a copper grid coated with amorphous carbon support. HRTEM images BaTiO3 NPs were recorded with a JEOL JEM ARM 200 F electron microscope (JEOL (Europe) SAS), which works at 200 keV.

2.4 X-ray diffraction investigations

X-ray diffraction (XRD) patterns of TPU, TPO, BaTiO3 and their composites were performed in a theta-theta configuration, in the angular range of 2θ=50 – 65°, with a Bruker D8 Advance diffractometer (Bruker, Hamburg, Germany).

2.5 FTIR spectroscopic investigations

The IR spectra of TPU, TPO, BaTiO3 and their composites were recorded using a FT-IR spectrophotometer, Vertex 80 model, from Bruker (Billerica, MA, USA), with a resolution of  2 cm-1.

2.6 FTRaman spectroscopic investigations

The Raman spectra of TPU, TPO, BaTiO3 and their composites were recorded with and a FT-Raman spectrophotometer, RFS 100S model, from Bruker (Ettlingen, Germany), with a resolution of 1 cm-1.

2.7 Photoluminescence investigations

The photoluminescence (PL) spectra of TPU, TPO, BaTiO3 and their composites were recorded with a Fluorolog-3 spectrophotometer, FL3-2.2.1 model, from Horiba Jobin Yvon (Palaiseau, France).

2.8 Dielectric properties

The dielectric properties of the composites based on the PTU:TPO blends and the BaTiO3 NPs were examined using dielectric spectroscopy (DS) with the equipment having a high resolution Alpha-A Analyzer, from NOVOCONTROL GmBH. The electrical properties were determind at room temperature, the frequency being between 0.01 Hz and 10 MHz and the alternating voltage having the value of 0.3 V. The samples are in the form of platelets with a diameter of approximately 13 mm their thicknesses varying between 0.095 mm and 0.160 mm. The samples are placed between two circular metal electrodes, in a capacitor configuration with plane-parallel armatures.

2.9 Differential scanning calorimetry (DSC) investigations

DSC measurements were performed with a DSC 204 F1 type from Netzsch (Selb, Germany). The experiments were conducted from room temperature to 500 °C in an inert atmosphere of helium (40 mil/min flow gaze rate) and a heating rate of 5°C/min. The samples were sealed in Al crucibles by pressing and as reference was used an empty Al crucible. The accuracy of heat flow measurements was ±0.001 mW.“ “

 Results: The results are very well described and detailed. 

The schematic of the polymers could be improved. It is very hard to see what is described in the text. Consider making a scheme with different colors or different thicknesses of bonds. 

I believe that the polymer schematic could be presented at the beginning of the discussion, for clarity. Also, a schematic showing the film and inorganic particles could make it easier to understand the systems synthsized. 

Authors reply: In the revised manuscript, different thicknesses of bonds are used in Scheme 1 and Scheme 2.

Minor Comments:

Line 50. low weight, high rigidity or good chemical resistance. The use of “or” makes the sentence a little bit confusing. If appropriate consider change it to “and”

Authors reply: In the revised manuscript, comment from line 50  is situated at line 55, this being  changed as follows:

“Thermoplastic polyolefin (TPO), known as polyethylene-polyoctene rubbers, is one of the most common and commercial thermoplastic polymer used due to its high rigidity and good chemical resistance [15, 16]. “

Line 53. Since the authors decide to use the date (2013), this affirmation needs a reference.

Authors reply: The reference which sustains the effort in the development of new composites based on the BaTiO3 nanoparticles, since 2013 is Reference 20, with the following coordination

  1. Fan, B. H.; Zha, J. W.; Wang, D. R.; Zhao, J.; Zhang, X. F.; Dang, Z. M. Preparation and dielectric behaviors of thermoplastic and thermosetting polymer nanocomposite films containing BaTiO3 nanoparticles with different diameters. Compos Sci Technol 2013, 80, 66-72. https://doi.org/10.1016/j.compscitech.2013.02.021.

Line 56. Typo: graphene

Authors reply: In the revised manuscript, this mistake has been corrected. Please, see:

-lines 57-60, the sentence  “Since 2013, a sustained effort has been achieved in the development of new composites based on the BaTiO3 nanoparticles, thermoplastic polymers of the type polyvinylidene fluoride and various carbon nanoparticles such as grapene [18] or carbon nanotubes [19] for applications in the field of 3D printing devices, health and textile fibers.”

has been rewritten as

„Since 2013, a continuous effort has been devoted in the development of new composites based on the BaTiO3 nanoparticles (NPs), thermoplastic polymers of the type polyvinylidene fluoride (PVDF) and various carbon NPs such as graphene [20] or carbon nanotubes [21] for applications in the field of 3D printing devices [3, 22], medical devices [21] and textile fibers [21].”

Line 57. The word “health” needs more description. Are the authors intending to say medical devices?

Authors reply: In the revised manuscript, the word “health” has been replaced with the word “medical devices“. Please, see the lines 61-65:

“Since 2013, a continuous effort has been devotedin the development of new composites based on the BaTiO3 nanoparticles (NPs), thermoplastic polymers of the type polyvinylidene fluoride (PVDF) and various carbon NPs such as graphene [20] or carbon nanotubes [21] for applications in the field of 3D printing devices [3, 22], medical devices [21] and textile fibers [21].“

Line 66. The hardener should be described (which chemical compound it is)

Authors reply: In the revised manuscript, the following comment has been included at lines 97-102:

„The thermoplastic elastomers, i.e. TPU and TPO (including its hardner which corresponds to polydimethylsiloxane, marketed under the name of Sylgard TM 186 Silicone elastomer curring agent), were purchased from Elastollan-BASF Chemical Company (Cleveland, OH, SUA). The BaTiO3 nanopowder, N,N'-dimethyl formamide (DMF) and C2H5OH were purchased from Sigma Aldrich (St. Louis, MO, USA )..”

Line 71. Replace the chemical formula for ethanol.

Authors reply: Line 71 in the revised manuscript is find as line 101. This has been changed as follows:

“The thermoplastic elastomers, i.e. TPU and TPO (including its hardner which corresponds to polydimethylsiloxane, marketed under the name of Sylgard TM 186 Silicone elastomer curring agent), were purchased from Elastollan-BASF Chemical Company (Cleveland, OH, SUA). The BaTiO3 nanopowder, N,N'-dimethyl formamide (DMF) and C2H5OH were purchased from Sigma Aldrich (St. Louis, MO, USA ). “

Line 80. Please keep consistence on the figures, for clarity: 0.05, 0.10, 0.23, 0.30

Authors reply: In the revised manuscript, in Figure 3 has been added the BaTiO3 weight added to TPU:TPO =2:1 thermoplastic polymers

Line 81. The symbol for the Celsius degree is wrong. Please replace the symbol (it is not a superscribed  zero or “o”)

Authors reply: In the revised manuscript, the above change has been carried out on line 109-110, as follows:

“After mixing them ultrasonically for 5 min, the resulting solution was placed in a Petri dish and dried for 2 hours at 100 °C.“

Line 140. Use appropriate degree symbol, not superscribed Zeros or O

Authors reply: Line 140 is find in the revised manuscript as line 224. This has been changed as follows:

“The XRD pattern of BaTiO3 showed in Figure 2a, reveals the crystalline tetragonal phase of BaTiO3 according to the standard ICCD database (PDF 04-016-2042), the main diffraction peaks apear at 2θ equal to 22.2°, 31.5°, 38.9°, 45.4°, 51° and 56.3°, these being assigned to the cystalline planes (001), (110), (111), (002), (210) and (211), respectivley. “

Line 160. Please keep the text impersonal. As expected

Authors reply: Line 160 is placed in the revised manuscript as line 256. According to above request, in the revised manuscript, line 256 is written as follows:

“As expected, the XRD pattern of the blend TPU:TPO 2: 1 (Figure 3d) contains all peaks of the two thermoplastic polymers. “

Round 2

Reviewer 2 Report

the manuscript has been significantly improved - it can be now accepted for publication in the current form

Author Response

In the revised manuscript, the English language and style were checked spelling, these being highlighted with track changes.

Reviewer 3 Report

Major Comments:

The authors fixed some of the issues and incorporated important information into the text. The obtained data is extensively analyzed and described. 

I would suggest the authors verify the presence of typos and grammar errors before submitting the final version of the text. For a reviewed file there are too many typos, which could be fixed by any text software (eg. word). If necessary, try to check the text on a computer with English as the default language. 

I still think that the chemical scheme could be improved. Probably with a stick representation, instead of having the description of -(CH2)-. It is very hard to understand the phenomenon explained in the text. I would suggest the use of Chemistry software like Chemdraw

Some spectra have a legend with the name of the compounds or blends. For clarity, I believe it would be better to add this information to all spectra. 

Minor Comments:

Line 74 – in situ – It should be in italic: in situ

100 – USA

114 -ca.

  1. Please keep consistency regarding significant figures. If using 43.4 the next number should be 44.0 and so on.

  1. working at 200 keV

  1. Please consider changing the word investigation for analysis, on all the cases.

  1. Please pay attention to the degree symbol

148. with and   a

  1. there is a typo: determind

  1. Please use the past tense. The samples were

  1. The symbol for milliliter is mL, not mil

  1. there is by the end of the sentence

  1. Such a structure

  1. Consider replacing the word mass. Suggestion: in the structure

  1. Do the authors mean smaller than 100 nm?

  1. Wrong degree symbol. Also in line 236 and 237

293 Thermoplastic

  1. The sentence sounds confusing. Please rephrase it.

  1. Returning

  1. Extrinsic

  1. Please replace the expression Ambiental conditions

  1. Enhancement
  2. Intrinsic

  1. The equation needs more space. Consider adding a line or two after the equation

  1. Increase

Table 1. Please keep consistency in using significant figures. 76.37 so 36.20 (if the case)

Author Response

I would suggest the authors verify the presence of typos and grammar errors before submitting the final version of the text. For a reviewed file there are too many typos, which could be fixed by any text software (eg. word). If necessary, try to check the text on a computer with English as the default language. 

Authors reply: In the revised manuscript, the English language and style were checked spelling, these being highlighted with track changes.

I still think that the chemical scheme could be improved. Probably with a stick representation, instead of having the description of -(CH2)-. It is very hard to understand the phenomenon explained in the text. I would suggest the use of Chemistry software like Chemdraw

Authors reply: In the revised manuscript, Scheme 1 and 2 have been rewritten in Chemdraw 4.1

Some spectra have a legend with the name of the compounds or blends. For clarity, I believe it would be better to add this information to all spectra. 

 Authors reply: In the revised manuscript, we have included a legend with the name of the compounds or blends in Figures 4, 5, 6, 7 and 8.

Minor Comments:

Line 74 – in situ – It should be in italic: in situ

Authors reply: In the revised manuscript, Line 74  is line 70. At line 70 “in situ“ is written in italic.

100 – USA

Authors reply: In the revised manuscript, line 100 is localized at line 96. At line 96 we have written USA.

114 -ca.

Authors reply: In the revised manuscript, line 114 is localized at line 111. At line 111, the “cca.” has been replaced with “ca.”

  • Please keep consistency regarding significant figures. If using 43.4 the next number should be 44.0 and so on.

Authors reply: In the revised manuscript, line 123 is localized at line 118 . At line 118 the sentence is rewritten as follows:

“As increasing the BaTiO3 NPs concentration in the polymers mass, from 6.25, 12, 25 and 30 wt.%, the thickness of films was changed to ~ 43.4, 44.0, 44.6 and 45 µm. “

  1. working at 200 keV

 Authors reply: In the revised manuscript, line 135 is localized at line 132. At line 130-132, the sentence has been rewritten as follows:

“HRTEM images BaTiO3 NPs were recorded with a JEOL JEM ARM 200 F electron microscope (JEOL (Europe) SAS), working at 200 keV. “

  1. Please consider changing the word investigation for analysis, on all the cases.

  Authors reply: In the revised manuscript, at lines 123, 134, 139, 144, 149 and 164, we have replaced the word “investigation” with “analysis”.

  1. Please pay attention to the degree symbol

Authors reply: In the revised manuscript, we have corrected the degree symbol at line 136, 228 and 229.

  1. with and  a

 Authors reply: In the revised manuscript, line 148 is localized at 145. At line 145, the sentence has been rewritten as follows:

“The Raman spectra of TPU, TPO, BaTiO3 and their composites were recorded with a FT-Raman spectrophotometer, RFS 100S model, from Bruker (Ettlingen, Germany), with a resolution of 1 cm-1. “

  1. there is a typo: determind

 Authors reply: In the revised manuscript, this typo has been corrected. Thus, line 158 has been rewritten as follows:

“The electrical properties were determined at room temperature, the frequency being between 0.01 Hz and 10 MHz and the alternating voltage having the value of 0.3 V. “

  1. Please use the past tense. The samples were

 Authors reply: In the revised manuscript, line 165 is localized at line 159. The sentence has been rewritten as follows:

“The samples were in the form of platelets with a diameter of approximately 13 mm and their thicknesses varying between 0.095 mm and 0.160 mm. “

  1. The symbol for milliliter is mL, not mil

Authors reply: In the revised manuscript, line 173 is localized at lines 166-167. We  have been rewritten the sentence as follows:

“The experiments were conducted from room temperature to 500 °C in an inert atmosphere of helium (40 mL/min flow gaze rate) and a heating rate of 5 °C/min.  “

  1. there is by the end of the sentence

 Authors reply: In the revised manuscript, the line 175 is localized at line 167. The sentence from lines 167-168 has been rewritten as follows:

“The samples were sealed by pressing and as reference was used an empty Al crucible. “

  1. Such a structure

 Authors reply: In the revised manuscript, line 188 is localized at lines 182-184; the sentence has been rewritten as follows:

“Such structure is described by a high and low glass transition temperature that induces the thermodynamic incompatibility between hard and soft segments. “

  1. Consider replacing the word mass. Suggestion: in the structure

 Authors reply: In the revised manuscript, we have replaced the word “mass” with “structure”. Please, see lines 193 and 279.

  1. Do the authors mean smaller than 100 nm?

 Authors reply: According to HRTEM analysis, the BaTiO3NPs have the mean size equal to 53.35 ± 9.89 nm.

In the revised manuscript, we have rewritten line 206 as follows:

“SEM image of BaTiO3 NPs with the mean size equal to 53.35 ± 9.89 nm (f). “

  1. Wrong degree symbol. Also in line 236 and 237

 Authors reply: In the revised manuscript, lines 227-230 have been rewritten as follows:

“In contrast with TPU, significant differences are observed for the XRD pattern of TPO in the 2θ angular domain between 40° and 65°. The presence of peaks from 2θ equal to 44.3° and 64.5° in the XRD pattern of TPO were also reported by Z.J. Zhang et al. [30].  “

293 Thermoplastic

Authors reply: In the revised manuscript, line 281-283 has been rewritten as follows:

“These changes suggest an adsorption of thermoplastic polymers on the surface of BaTiO3 NPs via C-H bonds of the TPU amide groups. “

  1. The sentence sounds confusing. Please rephrase it.

 Authors reply: In the revised manuscript, line 337 is localized at lines 325-326 ; the sentence has been rewritten as follows

“Such a variation can also be observed in Figure 6, when the massic ratio between the two thermoplastic polymers changes.  “

  1. Returning

 Authors reply: In the revised manuscript, line 362 is localized at line 351 and the sentence has been rewritten as follows:

“Returning at Figure 7, above variations indicate a chemical interaction between TPU and BaTiO3 NPs which can be described as an exchange reaction that takes place according to Scheme 2. “

  1. Extrinsic

 Authors reply: In the revised manuscript, line 386 is localized at lines 374-377; the sentence  has been rewritten as follows:

“However, a possible speculation of this result may be the material surface oxidation or the formation of extrinsic carbonyl groups at the TPO surface due to the recorded PL spectrum under normal conditions, namely in the presence of O2 and water vapours.  “

  1. Please replace the expression Ambiental conditions

 Authors reply: In the revised manuscript, lines 374-377 have been rewritten as follows:

“However, a possible speculation of this result may be the material surface oxidation or the formation of extrinsic carbonyl groups at the TPO surface due to the recorded PL spectrum under normal conditions, namely in the presence of O2 and water vapours.  “

  1. Enhancement

Authors reply: In the revised manuscript, lines 405-408  have been rewritten as follows:

“The following variations are highlighted in Figure 9: i) an enhancement in the intensity of PL spectra of the TPU:TPO 2:1, TPU:TPO 4:1 and TPU:TPO 6:1 blends of 1.25, 4 and 7 times higher than that of the TPU-TPO 1:1 and .... “

  1. Intrinsic

 Authors reply: In the revised manuscript, lines 444-447 have been rewritten as follows:

“Moreover, the π–π interactions of phenyl groups are connected with the carbonyle units coming from the intrinsic chemical structure of TPU, being stabilized and isolated by the surrounding polyolefins segments similar to a coiled membrane. “

  1. The equation needs more space. Consider adding a line or two after the equation

 Authors reply: In the revised manuscript, we have included a line after the equation (1).

  1. Increase

 Authors reply: In the revised manuscript, line 535 is localized at lines 522-524, these were rewritten as follows:

“The effect is to increase the characteristic time of the TPU:TPO 2:1 blend with the increase of BaTiO3 NPs concentration. “

Table 1. Please keep consistency in using significant figures. 76.37 so 36.20 (if the case)

 Authors reply: In the revised manuscript, Table 1 has been corrected according this comment. Please, see line 543 – 36.20, 38.70.

Bottom of Form